# Biostimulation of Anaerobic Digestion Using Iron Oxide Nanoparticles (IONPs) for Increasing Biogas Production from Cattle Manure

**DOI:** 10.3390/nano12030497

**Published:** 2022-01-31

**Authors:** Dilbag Singh, Kamla Malik, Meena Sindhu, Nisha Kumari, Vijaya Rani, Shikha Mehta, Karmal Malik, Poonam Ranga, Kashish Sharma, Neeru Dhull, Shweta Malik, Nisha Arya

**Affiliations:** 1Department of Microbiology, CCS Haryana Agricultural University, Hisar 125004, India; sdilbag460@gmail.com (D.S.); meenasindhu20@gmail.com (M.S.); shikhamicrobio@gmail.com (S.M.); rangamicro@yahoo.co.in (P.R.); kashishsharma7000@gmail.com (K.S.); dhullneeru11@gmail.com (N.D.); 2Department of Biochemistry, CCS Haryana Agricultural University, Hisar 125004, India; nishaahlawat211@gmail.com; 3Department of Farm Machinery and Power Engineering, CCS Haryana Agricultural University, Hisar 125004, India; vijayarani@yahoo.com; 4Department of Agronomy, CCS Haryana Agricultural University, Hisar 125004, India; karmalsingh@gmail.com (K.M.); malik.shweta54@gmail.com (S.M.); 5Department of Textile and Apparel Designing, CCS Haryana Agricultural University, Hisar 125004, India; nishasangwan@gmail.com

**Keywords:** biogas production, anaerobic digestion, methane, cattle manure, iron oxide nanoparticles, *Azadirachta indica*

## Abstract

The effect of synthesised IONPs employing a nontoxic leaf extract of *Azadirachta indica* as a reducing, capping, and stabilizing agent for increasing biogas and methane output from cattle manure during anaerobic digestion (AD) was investigated in this study. Furthermore, the UV-visible spectra examination of the synthesized nanoparticles revealed a high peak at 432 nm. Using a transmission electron microscope, the average particle size of IONPs observed was 30–80 nm, with irregular, ultra-small, semi-spherical shapes that were slightly aggregated and well-distributed. IONPs had a polydisparity index (PDI) of 219 nm and a zeta potential of −27.0 mV. A set of six bio-digesters were fabricated and tested to see how varying concentrations of IONPs (9, 12, 15, 18, and 21 mg/L) influenced biogas, methane output, and effluent chemical composition from AD at mesophilic temperatures (35 ± 2 °C). With 18 mg/L IONPs, the maximum specific biogas and methane production were 136.74 L/g of volatile solids (VS) and 64.5%, respectively, compared to the control (*p* < 0.05), which provided only 107.09 L/g and 51.4%, respectively. Biogas and methane production increased by 27.6% and 25.4%, respectively using 18 mg/L IONPs as compared to control. In all treatments, the pH of the effluent was increased, while total volatile fatty acids, total solids, volatile solids, organic carbon content, and dehydrogenase activity decreased. Total solid degradation was highest (43.1%) in cattle manure + 18 mg/L IONPs (T5). According to the results, the IONPs enhanced the yield of biogas and methane when compared with controls.

## 1. Introduction

Biogas production is a well-established bioconversion technology (anaerobic digestion) for obtaining energy from biomass, which is a potent, renewable, and eco-friendly fuel [1]. One of the most important processes for transforming organic waste into valuable products and renewable energy in the form of methane, carbon dioxide, and other trace elements is anaerobic digestion [2,3]. With benefits such as reduced organic pollution, energy recovery, and low operating costs, anaerobic digestion (AD) has become a popular approach for recycling organic wastes into energy [4]. The use of additives in the anaerobic digestion process has further enhanced biogas output. Recent studies have revealed that the addition of nanoparticles has been found to affect the anaerobic digestion process and enhance biogas. The fundamental process of methane production is behind the interspecies electron transfer (IET) mechanism [5]. In this context, green nanotechnology, which is grabbing the attention of the biofuels and bioenergy sectors by playing the role of enhancer, improves production technology. Nanoparticles synthesised using green technology are preferred because of their low cost and similar processing characteristics to other metallic nanoparticles [6]. Nanotechnology’s contributions to catalysts, enzymes, and microbial immobilization are a novel sector for biofuel and bioenergy production [7]. Due to current sustainability concerns, research on eco-friendly and green synthesis for the development of nanoparticles based on plant extracts has gained a lot of attention. Finally, biological materials (plants) can be synthesized cheaply and thus can function as reducing and capping agents [8,9]. The common plant neem (*Azadirachta indica)* is a member of the *Meliaceae* family and is plentiful in India. According to the literature [10,11,12], the major active phytochemicals of *Azadirachta indica* are azadirachtin, nimbin [13], nimbolinin, nimbidin [14], quercetin, gedunin [15], salannin, ascorbic acid, and amino acids [16]. The fresh leaf extract of neem also contains various active biological compounds such as terpenoids, sitosterol, polyphenolic flavonoids, saponins, and alkaloids that act as reducing and capping agents and aid in the stabilization of nanoparticles [10]. As a result, there is a rising need to develop nanoparticle synthesis techniques that are clean, non-toxic, and ecologically benign. There is still scope for progress in terms of inventive techniques to improve the biogas production system’s efficiency, productivity, and long-term viability. As examined and discussed by Meegoda et al. [1] and Tabatabaei et al. [17], anaerobic digestion is the main biological advancement for improving the biogas production process but further research is needed to increase the efficiency and reduce the associated cost and time. Therefore, it is critical to create green technologies for nanoparticle synthesis that are less harmful to human health and the environment. Recent studies have found that various types of IONPs increase the AD process [18,19] and the methane fraction in biogas [20,21]. In comparison to control, the addition of iron nanoparticles as an additive or catalyst to a co-digestion system significantly increased biogas output while lowering chemical oxygen consumption and stimulating microbial activity [22]. Despite the fact that scientists have attempted to produce biogas from a variety of lignocellulosic plant materials, few studies have been carried out to enhance biogas production using plant-derived nanoparticles. As a result, the influence of successfully synthesised and characterized IONPs on biogas generation, methane yield, and chemical composition during the anaerobic processes of cattle manure has been investigated.

## 2. Materials and Methods

### 2.1. Experimental Materials

The cattle manure was collected from the dairy farm of Lala Lajpat Rai University of Veterinary and Animal Sciences, Hisar, India. The main characteristics of the cattle manure, i.e., pH, TS, VS, TVFA, Organic carbon, Nitrogen, Carbon Nitrogen ratio, were analysed with the following analytical methods: Determination of pH: The pH of the cattle dung, the influents, and the effluents was determined using an Elico pH meter (Elico ltd., Hyderabad, India) and was 6.17.Determination of Total Solids (TS): 100 g sample was dried at 80 °C to a constant weight for determination of total solids. The residue left was weighed and taken as total solids. We arrived at a figure of 18.42% TS.Determination of Volatile Solids (VS): 1.0 g of fine dried powder sample was taken in a pre-weighed china crucible and incinerated in a muffle furnace at 550 °C for 1 h. The loss in weight of TS (total solids) to combustion was taken as volatile solids, which was found to be 82.0%.Determination of Organic Carbon Content: Organic carbon content of the samples was calculated as:
(1)$$\%C=\frac{\%\mathrm{Volatile}\;\mathrm{Solids}}{1.724}$$
Organic carbon was found to be 47.16%.Determination of cellulose, hemicellulose, and lignin: Cellulose, lignin, and hemicellulose contents were estimated by determining acid detergent fibre (ADF) and neutral detergent fibre (NDF) in the sample powder (AOAC, 2000) and were found to be 39.2, 16.5, and 11.82%, respectively.TVFA, Nitrogen, and C/N were analysed using standard protocols and were found to be 695 mg/kg, 1.32% (N) and 20.1 (C/N), respectively.

### 2.2. Chemicals

In this study, all the chemicals used were AR and GR grade (Sigma Aldrich, St. Louis, MO, USA and Himedia laboratories pvt. Ltd., Mumbai, India) and thus they did not need to be purified further.

### 2.3. Preparation of Neem (Azadirachta Indica) Leaves Extract

Young and healthy leaves of neem were collected from the Chaudhary Charan Singh Haryana Agricultural University campus, Hisar, India. The leaves were washed thoroughly with distilled water to eliminate dust particles and then the air dried at room temperature. The leaves were cut into small pieces and grind into a fine powder. In a conical flask, 5.0 g of finely ground neem leaf powder was added to 100 mL of sterile distilled water and boiled for 10 min at 80 °C. Then, the extracts were filtered twice through Whatman filter paper No. 1 to remove particulate matter and centrifuged at 5000 rpm for 5 min [11]. The green, clear filtered solution of neem leaf extract was stored at 4 °C for further use. Initially, the TS, VS, and total phenolics of neem leaf were 17.01%, 92.30%, and 27.02 (mg/g), respectively.

### 2.4. Biosynthesis of IONP

IONP nanoparticles were synthesized utilizing neem (Azadirachta indica) leaf extract as the green reducing agent using modified protocols [23,24]. Neem leaf extract and iron (III) chloride hexahydrate (FeCl_3_·6H_2_O) solution (0.03 M) were dissolved in 100 mL of sterile deionized water at a 1:1 ratio in a procedure. The resulting mixture was heated at 80 °C for 20 min, stirring occasionally with a magnetic stirrer, until the pale yellow colour became brownish black. The appearance of an instantaneous black colour indicated the synthesis of IONPs, and the reaction mixture was then allowed to cool to room temperature. A decrease in the pH of the solution was also observed, indicating the synthesis of IONPs. The solution was poured into a beaker after 30 min and the supernatant was discarded by magnetic decantation. The black precipitates of IONPs were again washed with 20 mL of distilled water after centrifugation at 10,000 rpm for 15 min. The supernatant was removed once again. The pellet was transferred to a vial and 10 mL of ethanol was added to remove excess phytochemicals and supernatant. After we obtained the black powder, it was freeze-dried overnight and used for further characterization.

### 2.5. Characterization of IONPs

A UV-visible Eppendorf Bio spectrometer (Systronics, Ahmadabad, India) between 300 and 700 nm with a resolution of 1.0 nm was used to characterize effectively produced IONPs. The particle size of the IONPs was determined using a laser light scattering nano-size particle analyser and a zeta sizer Nano ZS-90 (Malvern Instruments, Cambridge, UK). Transmission electron microscope (Hitachi H-7650-80 KV, Leipzig, Germany) scanning at 100 kV was used to characterize the size and shape of the synthesized IONPs. Drop deposition onto a carbon/formvar coated copper grid yielded approximately 10–20 L of samples, which were air dried before imaging with a microscope.

### 2.6. Experimental Setup

The experiments were performed to evaluate the effects of different concentrations of synthesized and characterized IONPs on biogas and methane content during the AD of cattle manure (Table 1).

Six laboratory scale batch experiments were set up with the capacity of 5 litre glass bottle digesters fed with cattle manure (3 kg) and IONPs at 35 ± 2 °C (Figure 1). 

The IONPs were ultrasonically treated in deionized water for 7 min prior to being added to batch type AD to prevent aggregation and increase their stability. Six glass bottles were flushed with nitrogen flow and sealed with a butyl rubber stopper to establish anaerobic conditions. The gas collecting bottle was filled with water and the gas produced in the digester displaced water in the collecting bottle, which helped with the quantitative analysis of biogas. The sampling was conducted using a syringe at different time intervals for pH, TS, VS, OC, and TVFA using standard methods [25]. The samples were also analysed for dehydrogenase activity as described by Casida et al. [26]. The biogas was collected in graduated test tubes and its volume was measured based on the water displacement method. Methane content was estimated by a portable biogas analyser for eight weeks of digestion.

### 2.7. Statistical Analysis

All tests were performed in triplicate, and the significance of the results was determined using analyses of variance (ANOVAs). Statistical significance was defined as *p* < 0.05.

## 3. Results and Discussion

### 3.1. Biosynthesis of IONPs 

The aqueous solution of neem leaf extract was mixed with iron (III) chloride hexahydrate (FeCl_3_·6H_2_O) solution. The precursors’ colour changed from pale yellow to light orange, then black, and the pH dropped from 6.50 to 2.59. (Figure 2). It was observed that the instantaneous black colour change takes approximately 30 min and after that no change in the reaction mixture’s colour was observed. The results were in agreement with Christensen et al. [27], where the colour changed from a slight yellowish to brown and finally to black. After 24 h, there was no more change in colour in the reaction mixture. This indicated that all the iron salts in the reaction solution had been completely reduced. The presence of IONPs synthesized by the reduction of iron salts led to the colour and pH of the solution changing. It was proposed that neem leaf extract acts as a reducing agent due to the presence of compounds like terpenoids and flavanones [28].

### 3.2. Characterization of IONPs

A UV-Vis spectral analysis was used to confirm the formation and stability of IONPs in aqueous solution (Figure 3). The excitation of molecules by surface plasmon vibrations in IONPs was attributed to the presence of a significant peak at 432 nm. Devatha et al. [29] found identical UV-VIS spectra for synthesised Fe_3_O_4_ NPs in their studies. Precipitated magnetite nanoparticles take on colour as a result of molecular excitation produced by surface plasmon resonance (SPR). Effective absorption occurs when the frequency of the electromagnetic field remains coherent with the electric motion. Aqueous colloidal solution has been used to confirm the synthesis and stability of IONPs. The maximum absorbance was read at 229 nm. There was no more absorption after 500 nm, resulting in a reduction and the formation of IONPs. A characteristic surface plasmon resonance band for IONPs was observed at a wavelength of 190–250 nm [30]. 

Turakhia et al. [31] reported identical UV-visible spectra for green synthesis of IONPs from *Spinacia oleracea* with a peak at 404 nm. The absorption spectrum of IONPs was observed at 371.71 nm when prepared via the green route using *Pomegranate granatum* seeds [23]. During this study, the IONPs were synthesized in 30 min using a green process involving non-toxic neem leaves, making this method suitable for practical application. Furthermore, a particle size analyser (PSA) was used to measure the size of IONPs and the polydispersity indexes (PDI) were observed to be 219 nm (Figure 4). The average particle size and the polydispersity index (PDI) indicate the qualities most closely related to the size distribution of nanoparticles.

The zeta potential (surface potential) has a direct relationship with the stability of nanoparticles and was −27.0 mV (Figure 5), indicating the good stability of the synthesised IONPs. Furthermore, the synthesized IONPs maintained their stability one month after synthesis. Due to the electrostatic repulsion of individual particles, a positive or negative zeta potential of nanoparticles indicates the strong physical stability of nanosuspensions. A zeta potential beyond the range of −30 mV to +30 mV is generally thought to have adequate repulsive force to achieve better physical colloidal stability. Due to the van der Waals attractive forces acting on nanoparticles, a small zeta potential value can cause particle aggregation and flocculation. It was observed that the zeta potential of IONPs with values of >+25 mV or 25 mV usually has a high degree of stability. Low dispersion ZP values will lead to aggregation, coagulation, or flocculation due to van der Waals interparticle attraction. The zeta potential is one of the key properties of the particle that can affect particle stability as well as its cell adhesion. Particle suspensions with positive or negative zeta potentials have a significant role in stabilizing them. This is attributed to the electrostatic repulsion between particles with the same electric charge that causes the segregation of particles [32].

TEM at 80 kV in high contrast imaging mode was used to examine the size, shape, and structure of the synthesized IONPs. Figure 6 shows TEM images taken from the drop coated grids of IONPs. It was observed that the synthesized IONPs were mostly irregular in shape and particle size, with some variations ranging from 30 to 80 nm, while the average mean particle size was around 52.5 nm.

Similar results for IONPs were investigated by Machado et al. [8] and Mystrioti et al. [33] using pomegranate, mulberry, and cherry leaf extracts. The average diameter of NPs was found to be 21.59 nm after 100 particles were selected to determine their size. The large, agglomerated clusters were formed on the basis of the accumulation of tiny building blocks of various bioactive reducing agents in the plant extract or as a result of the plant extract’s lower capping ability and the tendency of iron-based nanoparticles to agglomerate due to magnetic interactions. The IONPs were deeply surrounded by biological components from the neem leaf extract. Therefore, the resulting structure was a microstructure composed of IONPs and phytochemical compounds. 

### 3.3. Effect of IONPs on Anaerobic Digestion (AD) of Cattle Manure

During eight weeks of digestion, biogas and supernatant were measured daily to analyse the effect of IONPs on the AD of cattle manure until the biogas production stopped. Figure 7 shows the changes in accumulative biogas production. Six biodigesters started to produce biogas rapidly after the second week of digestion, and biogas yield increased by 27.6% in T5 compared to control (T1) after the fifth week. Compared with control (T1), IONPs increased biogas production in the initial stage but steadily decreased later on. T5 (18 mg/L) significantly increased accumulative biogas (29.48 L/g VS), followed by T6 (26.58 L/g VS) and T4 (25.0 L/g VS) compared with T1 (control), which was 22.91 L/g VS. 

The cumulative production of biogas recorded during anaerobic digestion is shown in Figure 8. The results show that the addition of IONPs in all concentrations enhanced the biogas production and shortened the lag phase. The cumulative production of biogas was found to be 136.74, 123.85, 117.83, 108.32, 102.62, and 107.09 l/g VS in the treatments T5, T6, T4, T3, and T1, respectively, after the eighth week of digestion. Similarly, the accumulative methane yield was highest in T5, i.e., 64.5% while that of the control was 51.4%. There was an increase in biogas (27.6%) and methane (25.4%) production as compared to control after eight weeks of batch anaerobic digestion in T5 at the laboratory scale. The highest biogas yield was obtained from 18 mg/L of IONPs, which corresponds to up to a 27.6% enhancement over the control bioreactor. Moreover, IONPs improved COD reduction efficiency to 42% [34]. According to Feng et al. [2], adding zerovalent iron (ZVI) accelerated the anaerobic digestion of sludge, increasing biogas and methane production. Methane production was boosted by increasing the ZVI dosage up to 20 g/L as compared to control, implying that these IONPs effectively stimulate microorganisms. The increases could have been attributed to the effect of IONPs in syntrophic/cooperative metabolism between electron-donating and electron-accepting microorganisms via direct interspecies electron transfer processes [35,36]. Yu et al. [37] studied the effect of zero valent iron (ZVI) on the anaerobic digestion of cow manure. With the addition of 10 g/L ZVI, the cumulative methane and VFA yields decreased by 10.3% and 12%, respectively. Previous research has shown that enhancing the performance of an anaerobic digester by adding zero-valent iron (ZVI) can boost the methane generation and degradation rate of organic wastes. The highest biogas production (400 mL/day) and methane yield (100% CH_4_) were attained with 2 g of Fe_2_O_4_-TiO_2_ MNPs as compared to the control biogas production (350 mL/day) and methane yield (65% CH_4_) from waste water [38]. IONPs improved the methanogenic process of cattle manure in the first few weeks, but they decreased biogas and methane yield in the following days due to the proliferation of slowly growing methanogenic bacteria and inhibition caused by decaying methanogenic bacteria.

Many nanoparticles (NPs) have been reported to improve biogas production from organic waste via anaerobic digestion (AD). However, the impact of IONPs on the stability of the AD process of cattle manure and the chemical composition of the effluent has yet to be thoroughly investigated. As a result, the maximum specific biogas production (*p* < 0.05) was achieved with T5 (18 mg/L Fe NP) with a value of 136.74 l/g VS and specific methane production enhanced by 64.5% (*p* < 0.05) as compared to cattle manure alone. 

### 3.4. Changes in Physicochemical Properties during Anaerobic Digestion of Cattle Manure

The anaerobic digestion of cattle dung supplemented with different treatments was tested for various parameters such as pH, TVFA, TS, VS, organic carbon, and dehydrogenase activity at the laboratory scale. The variations in pH during the anaerobic digestion of cattle dung are shown in Figure 9.

In the first two weeks, the pH value of each bio-digester rapidly decreased from 7.35 to 6.70, which may be due to the corrosion of IONPs generating OH^–^, thereby increasing the buffering capacity, which led to an increase in pH of all bio-digesters after eight weeks. T5 had the highest TVFA content (980 mg/kg) and T2 had the lowest (345 mg/kg) (Figure 10).

When the amount of TVFAs in the biogas was greater than 980 mg/kg, the digestion process was stopped. After treatment with IONPs, the total volatile fatty acid (TVFA) content did not vary significantly from that of the control. As shown in Figure 10, the TVFA ranged from 280 to 980 mg/kg, indicating that microbial activities were harmonized. The organic carbon varied from 42.52 to 48.14% in different treatments, and the maximum was found to be 48.14% in T2 (Figure 11). Maximum TS and VS were also found to be 19.15% and 87.0% in T2 (Table 2).

The increase was due to higher microbial activity caused by the availability of nutrients leading to greater degradation of solids and, consequently, higher biogas yield. Zupancic et al. [39] observed that a 10% increase in volatile suspended solids degradation efficiency was reported when household organic waste was co-digested with municipal sludge. The maximum degradation of TS (43.1%) and VS (7.5%) was observed in T5 and T6 after the 8th week of digestion (Figure 12), which is better as compared with earlier studies.

Changes in volatile solids content with the addition of IONPs resulted in increased organic matter decomposition, which contributed to the differences in methane outputs. When the cattle manure was treated with 18 mg/L IONPs (T5), the maximum TS degradation was seen, followed by 21 mg/L IONPs (T6), resulting in removal rates of 43.1 and 7.5%, respectively. These findings were in accordance with those of Abdelsalam et al. [19], who found that increasing the number of metallic nanoparticles added to cattle slurry bio-digesters and enhanced biogas and methane output. After the completion of eight weeks of batch anaerobic digestion of cattle manure, the pH was increased and other parameters (TVFA, TS, VS, OC) decreased in all treatments.

Dehydrogenase activity (microbial biomass) was also determined at 15-day intervals for eight weeks in a batch digestion manner. The dehydrogenase activity was increased from 1180 to 1436, 1275 to 1518, 1305 to 1687, 1295 to 1596, 1165 to 1344, and 1055 to 1335 g TPF/g sample/24 h up to the fourth week during batch digestion in treatments T1 to T6, respectively, and decreased afterwards. The maximum dehydrogenase activity was found in T3 (1305) and increased (1687) up to the fourth week, after which we observed a decrease in the dehydrogenase activity of all the treatments up to the eighth week (Figure 13).

After eight weeks of investigation, it was observed that the 18 mg/L IONPs (T5) treated bio-digester produced adequate amounts of biogas and methane when compared to the control, exhibiting good utilization of acetic, propionic, formic, and butyric acids. Because FeNPs accelerate the transfer of electrons from acetogens to methanogens as an energy source, the conversion of TVFAs to methane was increased. Noonari et al. [40] studied the influence of Fe_3_O_4_ nanoparticles on methane yield in the anaerobic co-digestion of canola straw and banana plant wastes with buffalo dung and found promising results.

## 4. Conclusions

This study reported on the successful use of *Azadirachta indica* as a green strategy for the synthesis of IONPs. As a result, IONPs were formed through the development of a black-coloured solution. The rapid reduction process confirmed the efficacy of *Azadirachta indica* extract as a reducing and stabilizing agent. The spherical shape of IONPs with an average size of 52.5 nm and an irregular structure was confirmed using TEM imaging. On the other hand, the impact of IONPs on the stability of the AD process of cattle manure and the chemical composition of the effluent was examined. Five different concentrations of IONPs were applied to a lab-scale batch bio-digester. The results revealed that cumulative biogas and methane yields with the addition of IONPs (18 mg/L) as a supplement reached 136.74 L/g VS and 64.5%, respectively, which were 27.6% (biogas) and 25.4% (methane) greater than the corresponding values of the control group after the eighth week of digestion.

## Figures and Tables

**Figure 1 nanomaterials-12-00497-f001:**
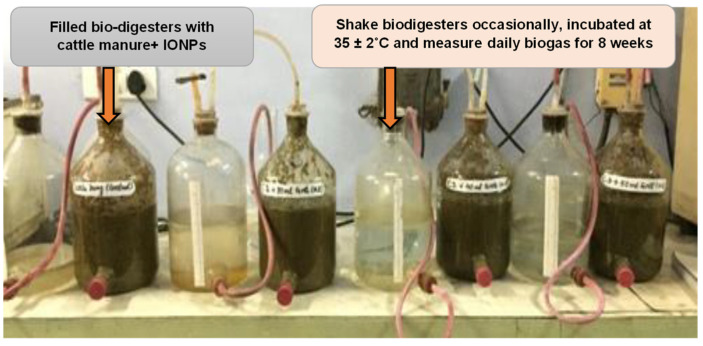
Schematic diagram of a laboratory-scale biodigester experimental setup for biogas production.

**Figure 2 nanomaterials-12-00497-f002:**
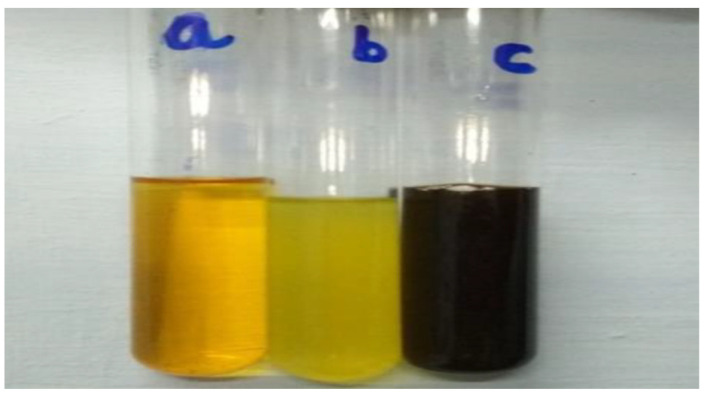
Synthesis of IONPs using *Azadirachta indica* (a—ferric chloride aqueous solution; b—neem leaf extract; c—synthesized IONPs).

**Figure 3 nanomaterials-12-00497-f003:**
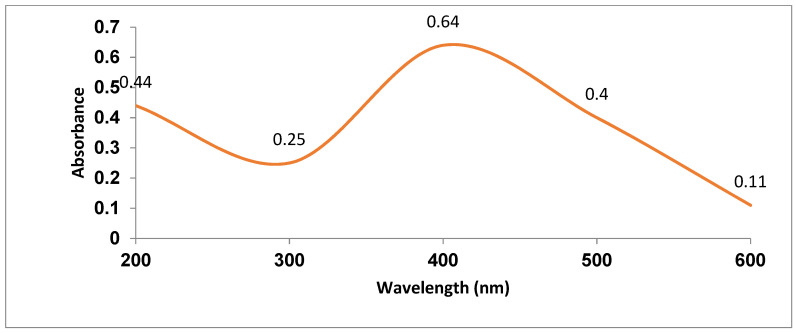
UV-visible absorption spectra of IONPs.

**Figure 4 nanomaterials-12-00497-f004:**
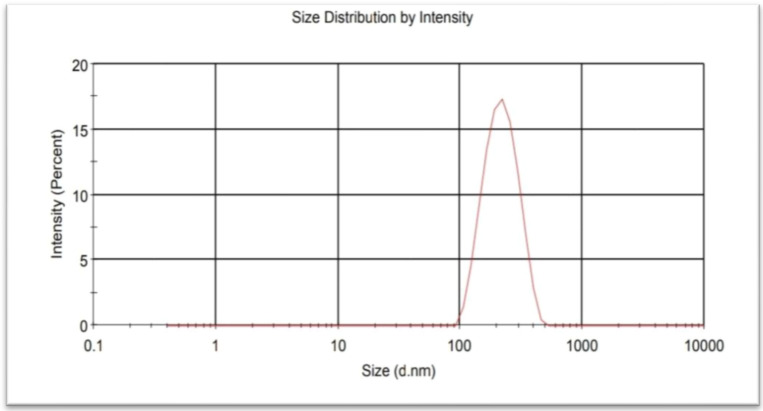
PSA of synthesized IONPs from neem leaf extract.

**Figure 5 nanomaterials-12-00497-f005:**
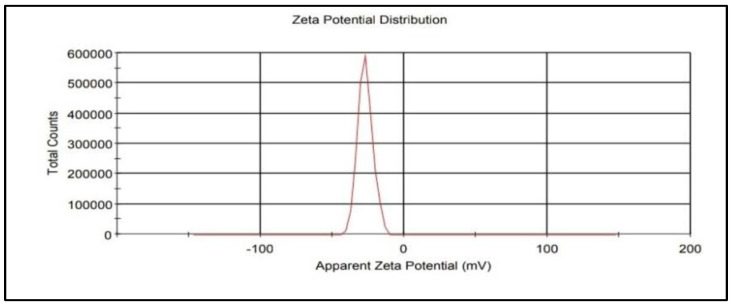
Determining the zeta potential of IONPs.

**Figure 6 nanomaterials-12-00497-f006:**
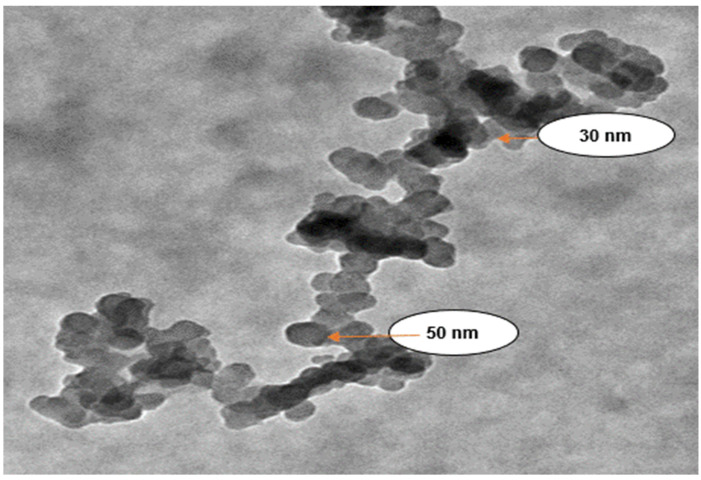
A TEM micrograph of synthesized IONPs.

**Figure 7 nanomaterials-12-00497-f007:**
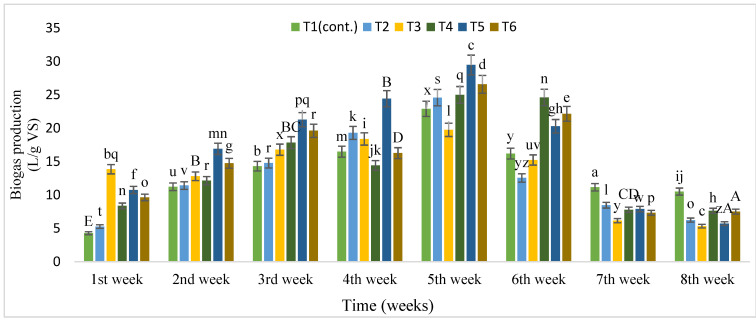
Biogas production (L/ g of VS) from cattle manure with different concentrations of IONPs under batch digestion. Bars with different letters indicate significantly different values at *p* ≤ 0.05.

**Figure 8 nanomaterials-12-00497-f008:**
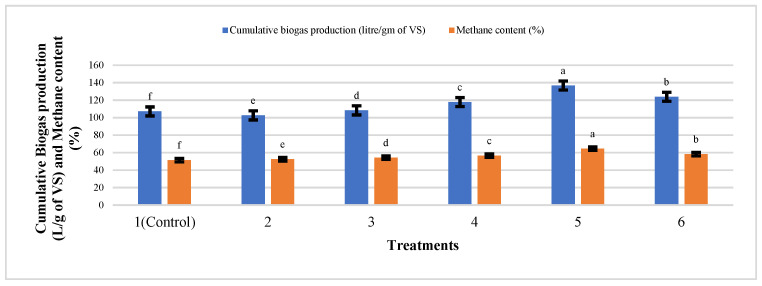
Cumulative biogas and methane production from cattle manure supplemented with different concentrations of IONPs. Bars with different letters depict significantly different values at *p* ≤ 0.05.

**Figure 9 nanomaterials-12-00497-f009:**
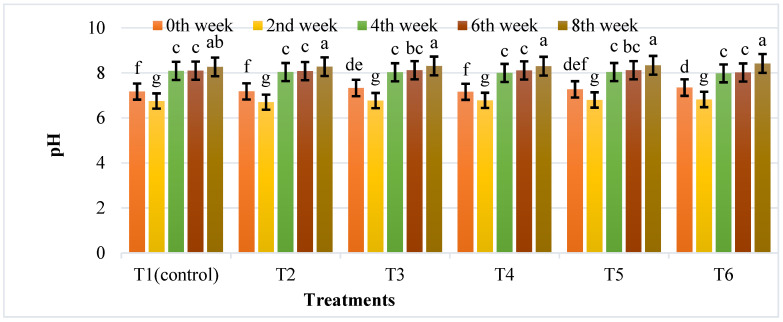
The effects of different IONPs treatments on the pH value of cattle manure during anaerobic digestion. Bars with different letters depict significantly different values at *p* ≤ 0.05.

**Figure 10 nanomaterials-12-00497-f010:**
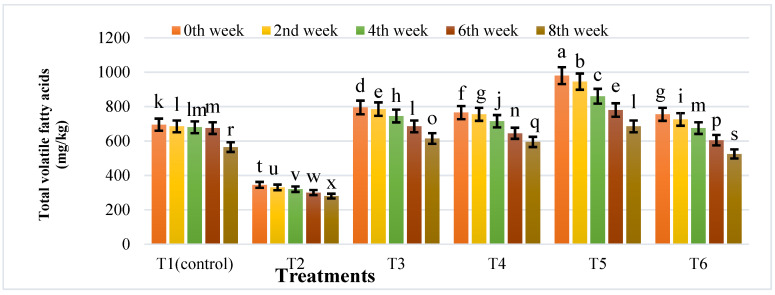
Effects of different treatments of IONPs on TVFA value during anaerobic digestion of cattle manure. Bars with different letters indicate significant differences (*p* ≤ 0.05) between treatments.

**Figure 11 nanomaterials-12-00497-f011:**
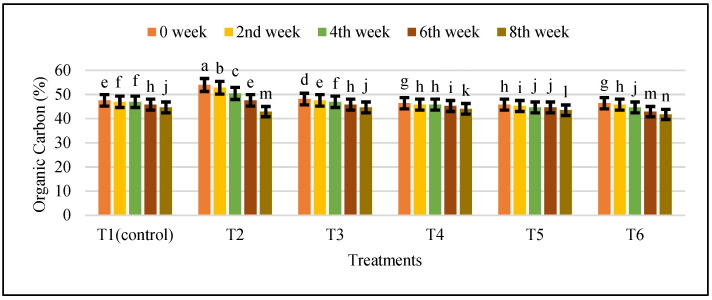
The effects of different treatments of IONPs on organic carbon content during anaerobic. digestion of cattle manure. Bars with different letters depict significantly different values at *p* ≤0.05.

**Figure 12 nanomaterials-12-00497-f012:**
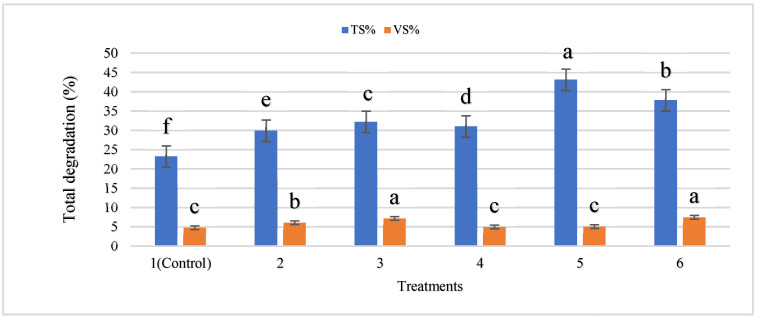
Degradation of total solids (TS) and volatile solids (VS) during anaerobic digestion. Bars with different letters depict significantly different values at *p* ≤ 0.05.

**Figure 13 nanomaterials-12-00497-f013:**
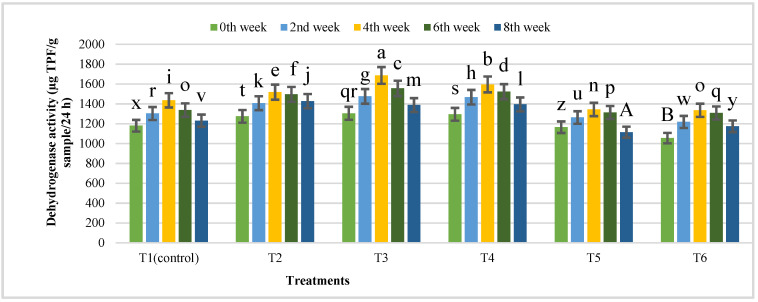
Dehydrogenase activity of different treatments of IONPs during the anaerobic digestion of cattle manure after eight weeks. Bars with different letters depict significantly different values at *p* ≤ 0.05.

**Table 1 nanomaterials-12-00497-t001:** Bio-digesters with different concentrations of IONPs for biogas production.

Treatments	Description
T1	Cattle manure (Control)
T2	Cattle manure + 9 mg/L IONPs
T3	Cattle manure + 12 mg/L IONPs
T4	Cattle manure + 15 mg/L IONPs
T5	Cattle manure + 18 mg/L IONPs
T6	Cattle manure + 21 mg/L IONPs

**Table 2 nanomaterials-12-00497-t002:** Effect of different treatments of IONPs on total solids and volatile solid degradation during eight weeks of batch digestion.

Time (Weeks)	1st		2nd		4th		6th		8th	
Treatments/ Parameters (%)	TS	VS	TS	VS	TS	VS	TS	VS	TS	VS
T1	18.42	82.0	17.80	81.0	16.90	80.0	15.4	79.0	14.15	78.0
T2	19.15	83.0	18.70	11.0	17.65	87.0	15.23	82.0	13.41	74.0
T3	18.73	82.3	17.88	82.0	16.25	81.0	14.10	80.0	12.70	77.0
T4	17.65	80.0	16.44	79.0	15.51	79.0	13.96	78.0	12.27	76.0
T5	17.83	79.0	16.77	78.0	15.03	77.0	12.50	77.0	10.14	75.0
T6	18.15	81.0	16.25	80.0	15.09	78.0	13.85	74.0	11.28	72.0

## Data Availability

The data used for the analysis in this study are available within the article, while the datasets used or analysed during the current study are available from the corresponding author upon reasonable request.

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
