# Peer review of "Biostimulation of Anaerobic Digestion Using Iron Oxide Nanoparticles (IONPs) for Increasing Biogas Production from Cattle Manure"

_nanomaterials, 2022, doi:10.3390/nano12030497_

Round 1

Reviewer 1 Report

I have gone through the ms. The presented work is worthy of publication. I have few minor comments:

  1. FENPs acronym be explained at the first instance
  2. In the title Bio stimulation are given separately, in my opinion, these should be together!
  3. Authors need to explicitly mention, how much % increase was there in biogas production (not in absolute figures);
  4. How there findings are better or comparable to similar other studies?
  5. Why is this technique better?
  6. What are the economic aspects of the technology they have proposed?

Author Response

Respected Reviewer

Reviewer 2 Report

January 11, 2022

Review of submission nanomaterials -1555410, entitled “Bio Stimulation of Anaerobic Digestion using Iron Nanoparticles for Increasing Biogas Production from Cattle Manure”

The submission is devoted to very important technological issue, but requires significant rewriting, since the authors use iron and iron oxide as interchangeable words which is incorrect. After correction it could be re-reviewed.

The authors need to clarify in the very title what are they studying.

In addition:

  1. In Key words Azadirachta indica should be in Italic: Azadirachta indica
  2. Manufacturers should be mentioned as (Brand (with the ownership type), City, Country) or if it is from USA – (Brand, City, State in 2 letters coding), not like this - “… (Eppendorf, USA) … (Nano ZS-90, Malvern Instruments, UK)... (TEM, Hitachi H-7650-80 KV, Germany) “. Please keep it consistent all over the submission.

Author Response

Respected Reviewer

Reviewer 3 Report

This paper is an interesting study focusing on the influence of synthesised and characterized iron oxide nanoparticles on biogas and methane generation during the anaerobic process of cattle manure.

The manuscript’s strengths are an integrated approach, in which Azadirachta indica was used in a green strategy for the synthesis of iron oxide nanoparticles. FeNP were synthesised, and characterized by UV, zeta sizer, laser light scattering nano-size particle analyser and TEM. The FeNP impact on the stability of the anaerobic digestion process of cattle manure and the chemical composition of the effluent was examined.

The weaknesses are that ecotoxicity of the final digestate after Fe-NP utilization wasn´t assessed which is a major weakness since manure can act as soil conditioners and provide valuable nutrients to plants as final stage utilization.  Because heavy metals are not biodegradable and can accumulate, reaching potentially toxic concentrations, toxicity of FeNP should have been evaluated.

Abstract – the information given is adequate and concise but short words (AD, TVFA, TS, VS) should be avoided (meaning is stated only in session 2.6, after several mentions throughout the manuscript).

  1. Introduction – The aim of paper is adequately exploited.
  2. Materials and methods

2.1 Explain what TS, VS and TVFA are.

Figure 1 – labels seem incomplete (Filled bio-digesters with and Shake biodigesters occasionally, incubated at…)

Figure 4 – I think you should rescale the x-axis scale, since all your NP are in the interval 100-1000 nm. Same comment on the figure 5.

How were pH, TS (total solids), VS (volatile solids), OC (organic carbon) and TVFA (total volatile fatty acids) determined? (equipment, name of method)

You state “A zeta potential beyond the range of 30 mV to +30 mV is generally thought to have adequate repulsive force to achieve better physical colloidal stability. Do you mean -30 to +30 mV?

Figure 7 and 8– label of y-axis – correct it (liters/gm of VS)

You use ZVI ( while increasing the ZVI dosage to 20 g/l boosted methane production as compared t ) before explaining what ZVI is (zero valent iron (ZVI) on the anaerobic digestion…)

Adequate discussion well documented by tables and figures but a rescaling of axis is urgent. Maybe too many figures, since with this scale some of them aren´t really useful but it´s just a suggestion.

You state “The highest biogas yield was obtained from 9 mg/L of iron NPs, which corresponds to up to a 37.6% enhancement over the control bioreactor.”, I cannot conclude that  that from figures 7 and 8, the highest shouldn´t be T5 (18 mg/L FeNP)?

Was the consortium of microorganisms studied?

Conclusions - Adequate

Sentence construction should be revised.

Eg . “As per examined and discussed by Meegoda et al. [1],Tabatabaei et al.[17] the mainline and downstream biological advancements for improving the biogas production process”

“When the cumulative production of biogas recorded during anaerobic digestion was shown in Figure 8.”

The paper is suitable for publication with major revisions.

Author Response

Respected Reviewer

Round 2

Reviewer 2 Report

Be consistent and use SI (see attachment)

Author Response

Respected reviewer

Reviewer 3 Report

Eventhough the majority of comments were taken into consideration and corrected by the authors there were three main questions that are still unanswered/uncorrected

Was the consortium of microorganisms studied?

y axis were not rescaled (Figures 4 and 5) nor the unit liters/gm was corrected (liters/g)(Figures 7 and8)

there is no comment on the ecotoxicity of the final digestate after Fe-NP utilization. Is that a concern, further studies should be assessed?

Author Response

Respected reviewer

"Please see the attachment
